

# Enhancement of stability of metastable states in the presence of Lévy noise

Alexander A. Dubkov[1], Claudio Guarcello[2,3] and Bernardo Spagnolo[4,5⋆]

**1** Radiophysics Department, Lobachevsky State University, 603950 Nizhniy Novgorod, Russia
**2** Dipartimento di Fisica "E. R. Caianiello", Università degli Studi di Salerno,
I-84084 Fisciano, Salerno, Italy
**3** INFN, Sezione di Napoli, Gruppo Collegato di Salerno –
Complesso Universitario di Monte S. Angelo, I-80126 Napoli, Italy
**4** Dipartimento di Fisica e Chimica "E. Segrè", Group of Interdisciplinary Theoretical Physics,
Università degli Studi di Palermo, I-90128 Palermo, Italy
**5** Stochastic Multistable Systems Laboratory, Lobachevsky University,
603950 Nizhniy Novgorod, Russia

⋆ bernardo.spagnolo@unipa.it

## Abstract

The barrier-crossing event for superdiffusion characterized by symmetric Lévy flights is analyzed. Starting from the fractional Fokker-Planck equation, we derive an integro-differential equation along with the necessary conditions to calculate the mean residence time of a particle within a fixed interval. We consider an arbitrary smooth potential profile, particularly metastable, with a sink and Lévy noise characterized by both an arbitrary index $\alpha$ and arbitrary noise intensity parameter. For the specific case of Lévy flights with an index $\alpha = 1$ and a cubic metastable potential, a closed expression for the mean residence time is obtained in quadratures. The analytical results reveal an enhancement of the mean residence time in the metastable state due to the influence of Lévy noise.



# 1 Introduction

Anomalous diffusion, which is a deviation from *normal* Gaussian diffusion, has one of the manifestations in Lévy flights. These are stochastic processes characterized by the occurrence of extremely long jumps, obeying the Lévy stable distribution. Lévy flights, characterized by a scale invariance property, are extensively observed in physics, chemistry, biology, ecological and financial systems, see [1–5] and references therein. Furthermore, using the Markovian property of Lévy flights, the generalized Kolmogorov equation can be derived from the Lévy noise-driven Langevin equation [6].

Metastability, as well as the transition process between metastable states, is a ubiquitous phenomenon in nature affecting different fields of natural sciences and advancing in its understanding is a key challenge in complex systems [7–28]. Experimental [24, 29–32] and theoretical [27, 33–36] results show that long-lived metastable states, even if observed in different areas of physics, were not fully explained. Furthermore, several studies have shown that the average escape time from a metastable state exhibits nonmonotonic behavior, peaking at a certain noise intensity, in systems governed by Gaussian diffusion, see [7–15] and references therein. This resonancelike behavior, which contrasts with the monotonic predictions of Kramers' theory [37], is known as the noise-enhanced stability (NES) phenomenon.

In this context, noise can actually enhance the stability of metastable states, leading to an average lifetime that exceeds the deterministic decay time. This raises an important question: what happens when a Brownian particle in a barrier-crossing process is replaced by a particle undergoing Lévy flights?

Recently, the first passage time properties of Lévy flights for random search strategies have been investigated [38–40]. Furthermore, noise-induced escape from a metastable state in the presence of Lévy noise governs a plethora of transition phenomena in complex systems, of physical, chemical and biological nature, ranging from the motion of molecules to climate signals, see [3, 20, 21, 23, 41–45] and references therein. The main focus in these papers is to understand how the barrier crossing in different potential profiles $V(x)$, is modified by the presence of the Lévy-stable noise $L_\alpha(t)$ with index $\alpha$. This is achieved by particle displacement analysis, which obeys the following Langevin equation

$$\frac{dx}{dt} = -V'(x) + L_\alpha(t),\tag{1}$$

where $\alpha$ is the stability index of the Lévy distribution, with $0 < \alpha < 2$. The main tools to investigate the barrier crossing problem for Lévy flights in these above-mentioned papers are the first passage times and residence times.

The problem of escape from metastable states driven by Lévy noise has garnered significant theoretical interest over the past two decades, with extensive research conducted through both numerical simulations and analytical approximations [3, 41, 44, 46–55]. In particular, rigorous mathematical results, in asymptotics, on the dominant scaling of the escape time of an overdamped Lévy-driven particle in a confined potential and in the weak noise regime have been obtained in Refs. [50, 51]. The backbone of the rigorous proof for deriving the asymptotic behavior of the escape time lies in decomposing the driving Lévy process into two components: one dominated by large jumps and the other by small jumps. The main result found by the authors was that the escape time from the attraction domains by Lévy jumps is always faster than that induced by Gaussian noise [50, 51]. The asymptotics of the escape rate was also studied mathematically in Ref. [46], in the context of paleo-climatic modelling. There, the authors found that the statistics of noise-induced jumping between metastable states in a potential is different for $\alpha$-stable noise from the usual Gaussian noise case. Furthermore, the stationary probability distribution deviates from the Gibbs distribution, and the waiting time for jumping depends in some cases more on the width than on the height of the barrier. In Refs. [3, 41, 48],

the barrier crossing process by a particle executing Lévy flights for three different types of potentials, namely bistable, metastable, and truncated harmonic potential, has been numerically investigated. Among the main results, the authors discovered a power-law dependence of the mean escape time on the noise intensity parameter over a wide range of values. Furthermore, for Cauchy noise, $\alpha = 1$, the authors develop the kinetic theory of the escape over the barrier in a bistable potential within the stationary flux approximation, assuming that the probability current across the barrier is constant. This is equivalent to requiring that the barrier is high in comparison to the noise intensity parameter. The authors found analytically the expression for the mean escape time.

Recently, in Ref. [44], the authors analyzed non-Gaussian escape rates, in particular Lévy flights, using a path integral framework and considering a weak-noise regime. The typical path is obtained by minimizing a stochastic action. The authors found that non-Gaussian noise always leads to more efficient escapes and can enhances escape rates by many orders of magnitude compared with thermal noise due to escape paths dominated by large jumps. The framework proposed in [44] allows to recover rigorous mathematical results on the dominant scaling of the escape rate for non-Gaussian noise in weak-noise regime [50,51].

However, despite its importance, exact analytical results for barrier crossing problems in metastable systems in the presence of Lévy noise remain elusive, making it an ongoing challenge in the field. This paper aims to answer this question by studying the barrier crossing process in a system with a metastable state driven by Lévy noise without any approximation and in particular for any value of noise intensity parameter and arbitrary index $\alpha$. Starting from the fractional Fokker-Planck equation corresponding to Eq. (1), we investigate the barrier crossing event by focusing on the mean residence time (MRT). Specifically, we analyze the average time a particle spends in the metastable state of the potential profile (see Fig. 1), which indicates its stability.

Here, we address the following open questions: (*i*) the exact results of the MRT of a particle moving in an arbitrary smooth potential profile with a sink, under the influence of Lévy noise with arbitrary index $\alpha$ and noise intensity parameter; (*ii*) a closed expression in quadratures for the MRT in the case of Lévy flights with index $\alpha = 1$ (Cauchy noise) in a cubic metastable potential; and (*iii*) the analytically derived enhancement of the stability of metastable states due to Lévy noise.

## 2 The model

The anomalous diffusion in the form of Lévy flights, for a particle moving in a potential profile $V(x)$, is described by the following fractional Fokker-Planck equation

$$\frac{\partial P}{\partial t} = \frac{\partial}{\partial x}\left[V'(x)P\right] + D_\alpha \frac{\partial^\alpha P}{\partial |x|^\alpha}, \tag{2}$$

where $P(x, t|x_0, 0)$ is the transition probability density, and $D_\alpha$ is the noise intensity parameter, in the sense that the size of a cloud of particles undergoing Lévy motion increases with time as $(D_\alpha t)^{1/\alpha}$. Here $\partial^\alpha/\partial |x|^\alpha$ is the Riesz fractional space derivative [4,56].

Equation (2) can be derived from different theoretical approaches [56–60], and in particular can be easily obtained directly from Eq. (1) [6]. Specifically, in this last paper, by exploiting the properties of random variables with infinitely divisible distributions [2,4,6], the characteristic functional of non-Gaussian white noise was obtained. Then, by applying a functional approach to decouple the correlation between stochastic functionals, the general Kolmogorov's equation for nonlinear systems driven by a non-Gaussian white noise source was derived. From this general equation we can obtain the fractional Fokker-Planck equation (2) for a Lévy-stable noise source $L_\alpha(t)$.

According to the definition, if the random process $x(t)$ initially starts from the value $x_0$ at $t = 0$, the residence time $T(x_0)$ in the given interval $(L_1, L_2)$ for the infinite observation time reads [43])

$$T(x_0) = \int_0^\infty \mathbb{1}_{(L_1, L_2)}(x(t)) \, dt, \tag{3}$$

where

$$\mathbb{1}_{(L_1, L_2)}(y) = \begin{cases} 1, & y \in [L_1, L_2], \\ 0, & \text{otherwise.} \end{cases} \tag{4}$$

Averaging Eq. (3), we find the mean residence time in the interval $(L_1, L_2)$

$$\tau_{MRT} = \langle T(x_0) \rangle = \int_0^\infty dt \int_{L_1}^{L_2} P(x, t | x_0, 0) \, dx. \tag{5}$$

The MRT is equivalent to the nonlinear relaxation time for diffusion in a potential profile with a sink (see Fig. 1), which was first defined in [61]. Subsequently, it was explored in arbitrary potential profiles and expressed in quadrature for Markovian processes in [62–66]. Changing the order of integration in Eq. (5), we arrive at

$$\tau_{MRT}(x_0) = \int_{L_1}^{L_2} Z(x, x_0) \, dx, \tag{6}$$

where

$$Z(x, x_0) = \int_0^\infty P(x, t | x_0, 0) \, dt. \tag{7}$$

Integrating Eq. (2) with respect to $t$ from 0 to $\infty$ and taking into account the initial condition $P(x, 0 | x_0, 0) = \delta(x - x_0)$ and the asymptotic condition $P(x, \infty | x_0, 0) = 0$ (for a potential with a sink), we obtain the following integro-differential equation for the function $Z(x, x_0)$

$$\frac{d}{dx}\left[V'(x) Z\right] + D_\alpha \frac{d^\alpha Z}{d|x|^\alpha} = -\delta(x - x_0). \tag{8}$$

To solve Eq. (8) it is better to consider the Fourier transform of the function $Z(x, x_0)$, i.e.,

$$\widetilde{Z}(k, x_0) = \int_{-\infty}^\infty Z(x, x_0) \, e^{ikx} \, dx. \tag{9}$$

For a smooth potential profiles $V(x)$, after Fourier transform, Eq. (8) can be written in the differential form

$$ik V'\left(-i\frac{d}{dk}\right)\widetilde{Z} + D_\alpha |k|^\alpha \widetilde{Z} = e^{ikx_0}. \tag{10}$$

It is convenient to introduce a new function $G(k, x_0)$, namely, the derivative of the function $\widetilde{Z}(k, x_0)$ with respect to $x_0$

$$G(k, x_0) = \frac{\partial}{\partial x_0} \widetilde{Z}(k, x_0). \tag{11}$$

Differentiating both parts of Eq. (10) with respect to $x_0$ we find

$$V'\left(-i\frac{d}{dk}\right)G - iD_\alpha |k|^{\alpha-1} \operatorname{sgn} k \, G = e^{ikx_0}, \tag{12}$$

where $\operatorname{sgn} x$ is the sign function.

Substituting $Z(x, x_0)$ from the backward Fourier transformation into Eq. (6) and changing the order of integration, we have

$$\tau_{MRT}(x_0) = \frac{1}{2\pi} \int_{-\infty}^{\infty} \widetilde{Z}(k, x_0) \frac{e^{-ikL_1} - e^{-ikL_2}}{ik} \, dk \,. \tag{13}$$

After differentiation of both sides of Eq. (13) with respect to $x_0$, in accordance with Eq. (11), we find

$$\tau'_{MRT}(x_0) = \frac{1}{2\pi} \int_{-\infty}^{\infty} G(k, x_0) \frac{e^{-ikL_1} - e^{-ikL_2}}{ik} \, dk \,. \tag{14}$$

One can easily check that, after replacing $k$ with $-k$, Eq. (12) coincides with the equation for the complex conjugate function $G^*(k, x_0)$, i.e. $G(-k, x_0) = G^*(k, x_0)$. As a result, Eq. (14) can be rearranged into a simpler form

$$\tau'_{MRT}(x_0) = \int_0^{\infty} \text{Re} \left\{ G(k, x_0) \frac{e^{-ikL_1} - e^{-ikL_2}}{\pi i k} \right\} dk \,, \tag{15}$$

where $\text{Re}\{...\}$ denotes the real part of the expression.

If a sink of the potential profile $V(x)$ is located at the point $x = \infty$ we have

$$\lim_{x_0 \to \infty} \tau_{MRT}(x_0) = 0 \,. \tag{16}$$

After integrating Eq. (15) with respect to $x_0$ and taking into account the condition (16), we find

$$\tau_{MRT}(x_0) = \int_{x_0}^{\infty} \text{Re} \left\{ \int_0^{\infty} G(k, z) \frac{e^{-ikL_2} - e^{-ikL_1}}{\pi i k} \, dk \right\} dz \,. \tag{17}$$

Thus, it is sufficient to solve Eq. (12) only in the region $k > 0$, obtaining

$$V'\left(-i \frac{d}{dk}\right) G - i D_\alpha k^{\alpha-1} G = e^{ikx_0} \,. \tag{18}$$

Equations (17) and (18), which are among the main results of the paper, give the exact relations useful to calculate the MRT of the symmetric Lévy flights with arbitrary index $\alpha$ and noise intensity parameter $D_\alpha$ in a smooth potential profile with a sink at $x = \infty$.

## 3 Results

**Metastable state and noise enhanced stability**  Now, we focus on a particle moving in a metastable cubic potential profile (see Fig. 1)

$$V(x) = -\frac{x^3}{3} + m^2 x \,, \tag{19}$$

and driven by a Cauchy-stable noise with Lévy index $\alpha = 1$. Here $x = m = x_{max}$ corresponds to the unstable equilibrium state, and $x = -m = x_{min}$ to the metastable state, with $m > 0$ any positive real number. Using Eq. (19) and placing $\alpha = 1$ in Eq. (12), we get

$$\frac{d^2 G}{dk^2} + \left(m^2 - i D_1 \, \text{sgn} \, k\right) G = e^{ikx_0} \,, \tag{20}$$

which for $k > 0$ becomes

$$\frac{d^2 G}{dk^2} + \left(m^2 - i D_1\right) G = e^{ikx_0} \,. \tag{21}$$

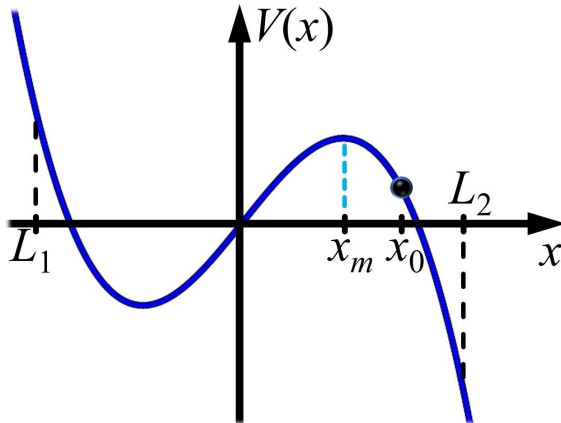

Figure 1: Cubic potential $V(x)$ with metastable state at $x = -m$, archetype model for any metastable state. $L_1$ and $L_2$ are the interval boundaries, $x_0$ is the initial position of the particle, $x_m = m$ is a potential parameter.

The general solution of Eq. (21) is the sum of the general solution of the homogeneous equation and its particular solution. Under the condition of its limitation, not divergent for arbitrary $k$, it takes the form

$$G(k, x_0) = C e^{-\lambda k} + \frac{e^{ikx_0}}{m^2 - x_0^2 - iD_1}, \tag{22}$$

where $C$ is an unknown complex constant and $\lambda$ is one of the complex roots

$$z = \pm\sqrt{iD_1 - m^2},$$

having a positive real part, $\lambda = \lambda_1 + i\lambda_2$, where

$$\lambda_1 = \left(m^4 + D_1^2\right)^{1/4} \sin\left[\frac{1}{2} \arctan\left(\frac{D_1}{m^2}\right)\right],$$
$$\lambda_2 = \left(m^4 + D_1^2\right)^{1/4} \cos\left[\frac{1}{2} \arctan\left(\frac{D_1}{m^2}\right)\right]. \tag{23}$$

To find the unknown constant $C$ we use the continuity conditions for the function $G(k, x_0)$ and its first derivative at the point $k = 0$

$$\lim_{k \to 0^+} G(k, x_0) = \lim_{k \to 0^-} G(k, x_0),$$
$$\lim_{k \to 0^+} \frac{dG(k, x_0)}{dk} = \lim_{k \to 0^-} \frac{dG(k, x_0)}{dk}. \tag{24}$$

For $k < 0$, Eq. (20) transforms into

$$\frac{d^2G}{dk^2} + \left(m^2 + iD_1\right)G = e^{ikx_0}, \tag{25}$$

and its solution under the condition of its limitation reads

$$G(k, x_0) = C^* e^{\lambda^* k} + \frac{e^{ikx_0}}{m^2 - x_0^2 + iD_1}. \tag{26}$$

Using Eqs. (22), (26) and conditions (24) we get

$$C + \frac{1}{m^2 - x_0^2 - iD_1} = C^* + \frac{1}{m^2 - x_0^2 + iD_1} \,,$$

$$-\lambda C + \frac{ix_0}{m^2 - x_0^2 - iD_1} = \lambda^* C^* + \frac{ix_0}{m^2 - x_0^2 + iD_1} \,. \tag{27}$$

The final expression for the constant $C$, obtained from the system (27), reads

$$C = -\frac{D_1 (x_0 + \lambda_2 + i\lambda_1)}{\lambda_1 [(m^2 - x_0^2)^2 + D_1^2]} \,. \tag{28}$$

Substituting Eqs. (22) and (28) into Eq. (17) and calculating the internal integral we arrive at

$$\tau_{MRT}(x_0) = \frac{D_1}{\pi} \int_{x_0}^{\infty} \left[ A \left( \frac{z + \lambda_2}{\lambda_1} \right) + B \right] \frac{dz}{(z^2 - m^2)^2 + D_1^2}$$

$$+ \frac{D_1}{\pi} \int_{x_0}^{\infty} \ln \left| \frac{z - L_1}{z - L_2} \right| \frac{dz}{(z^2 - m^2)^2 + D_1^2} + \int_{x_0}^{L_2} \frac{(z^2 - m^2) \, dz}{(z^2 - m^2)^2 + D_1^2} \,, \tag{29}$$

where

$$A = \arctan \frac{\lambda_2 + L_2}{\lambda_1} - \arctan \frac{\lambda_2 + L_1}{\lambda_1} \,,$$

$$B = \frac{1}{2} \ln \frac{\lambda_1^2 + (L_2 + \lambda_2)^2}{\lambda_1^2 + (L_1 + \lambda_2)^2} \,. \tag{30}$$

The exact quadrature formula of Eq. (29) is the other main result of the paper. By setting $D_\alpha = 0$ in Eq. (29), the dynamical time $\tau_d(x_0)$ is then obtained

$$\tau_d = \int_{x_0}^{L_2} \frac{dz}{z^2 - m^2} \,. \tag{31}$$

For $x_0 < m < L_2$, that is unstable initial conditions within the basin of attraction, the integral in Eq. (31) diverges, which means the impossibility for a particle to cross the potential barrier, located at the point $x = m$, in the absence of driving noise. For $m < x_0 < L_2$ we obtain the finite dynamical time

$$\tau_d(x_0) = \frac{1}{2m} \ln \frac{(L_2 - m)(x_0 + m)}{(L_2 + m)(x_0 - m)} \,. \tag{32}$$

For unstable initial conditions of the particle beyond the potential barrier (at $x = +m$) and within the interval $+m < x_0 < L_2$, the normalized MRT in the metastable state $\tau_{MRT}(x_0)/\tau_d(x_0)$ as a function of the noise intensity parameter $D_1$ has a nonmonotonic behaviour with a maximum (see curves in Figs. 2 and 3).[1] This is the noise enhanced stability (NES) phenomenon, already investigated with Gaussian noise sources [7–15, 17, 19, 20, 27, 28].

The normalized MRT $\tau_{MRT}(x_0)/\tau_d(x_0)$ for a metastable cubic potential as a function of the noise intensity parameter $D_1$ for different positions $L_1$ of the left boundary and a fixed value of the right boundary $L_2 = 4$ in a semilog plot is shown in Fig. 2. The different values of $L_1$ range from 0 to $-6$ with steps 0.2. A nonmonotonic behavior of the normalized MRT with a maximum as a function of the noise intensity parameter is observed for all values of $L_1$ analyzed, that is the particle is temporarily trapped in the metastable state.

---

[1]The MRT in the metastable state $\tau_{MRT}(x_0)$ as a function of the noise intensity parameter $D_1$, with fixed $L_2$, has the same nonmonotonic behaviour with a maximum but with different scaling in the vertical axis of Fig. 2. In Fig. 3 the MRT $\tau_{MRT}(x_0)$ versus $D_1$, with fixed $L_1$, is shown.

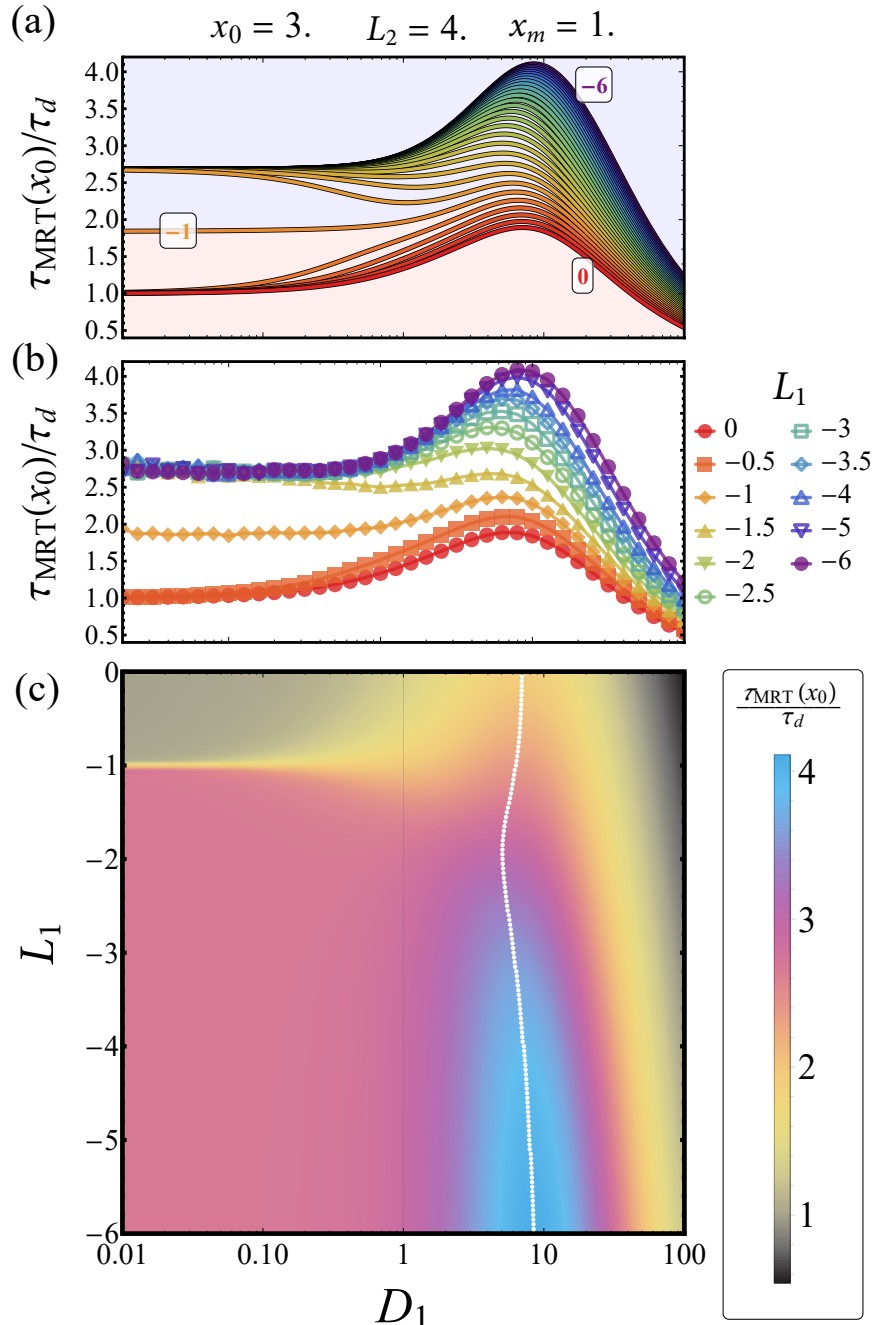

Figure 2: The results of analytical calculations, panels (a) and (c), according to Eq. (29), and numerical integration, panel (b), of the Langevin equation (1) for Cauchy noise, $\alpha = 1$. (a) Normalized MRT $\tau_{MRT}(x_0)/\tau_d$, from Eq. (29), for a metastable cubic potential as a function of the noise intensity parameter $D_1$ for different positions $L_1$ of the left boundary ranging from 0 to −6 with steps 0.2. (b) Numerical simulations of the Eq. (1) for the same quantity $\tau_{MRT}/\tau_d$ versus the noise intensity parameter $D_1$. (c) Density plot of $\tau_{MRT}(x_0)/\tau_d(x_0)$ versus $L_1$ and $D_1$ from Eq. (29). The white dotted line marks the position of the NES maxima. The parameter values are: $x_0 = 3.0$, $L_2 = 4$, $m = 1$.

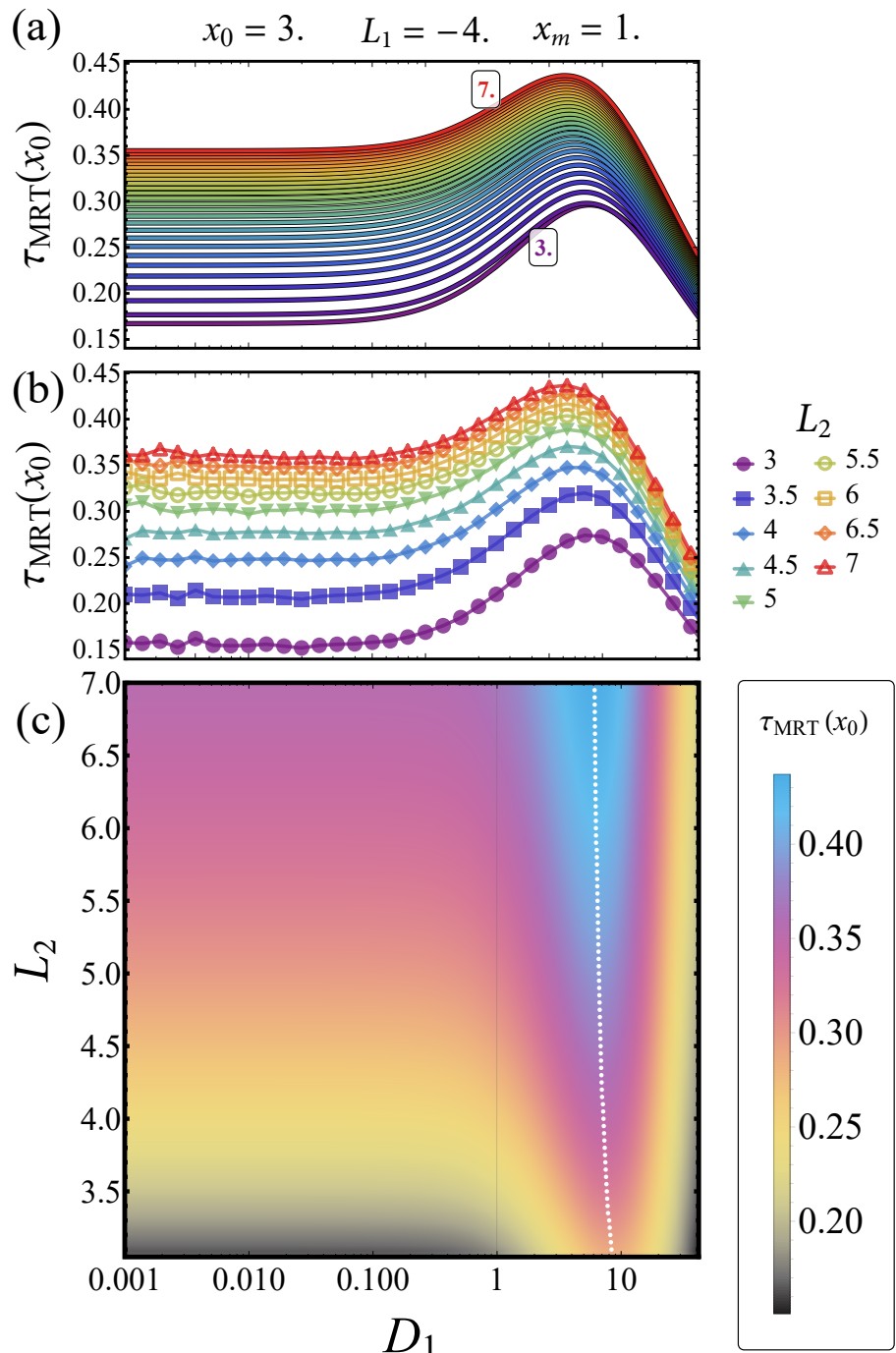

Figure 3: The results of analytical calculations, panels (a) and (c), according to Eq. (29), and numerical integration, panel (b), of the Langevin equation (1) for Cauchy noise, $\alpha = 1$. (a) MRT $\tau_{MRT}(x_0)$, from Eq. (29), for a metastable cubic potential as a function of the noise intensity parameter $D_1$ for different positions $L_2$ of the right boundary ranging from 3 to 7 with steps 0.2. (b) Numerical simulations of the Eq. (1) for the same quantity $\tau_{MRT}$ versus the noise intensity parameter $D_1$. (c) Density plot of $\tau_{MRT}(x_0)$ versus $L_2$ and $D_1$ from Eq. (29). The white dotted line marks the position of the NES maxima. The parameter values are: $x_0 = 3.0$, $L_1 = -4$, $m = 1$.

Furthermore, we perform the numerical integration of the Langevin equation (1) with the cubic potential profile of Eq. (19) for different position $L_1$ and fixed $L_2$ (see Fig. 2b), and different position $L_2$ and fixed $L_1$ (see Fig. 3b), across a wide range of noise intensity parameters $D_1$. The MRT is obtained by numerically integrating Eq. (1) over $2 \times 10^6$ time steps of width $dt = 10^{-3}$ and averaging over $5 \times 10^6$ independent numerical repetitions. The algorithm used for the Lévy noise source is that proposed by Weron for the implementation of the Chambers method, see Ref. [43].[2]

From these simulations we obtain the detailed dependence of the MRT on both the noise intensity parameter $D_1$ and the parameters $L_1$ or $L_2$. The agreement between the theoretical exact results of Eq. (29) and the numerical simulations of Eq. (1) is excellent.

For large noise intensity parameter, we have a power law behavior of the MRT as a function of the noise intensity parameter $\tau_{MRT}(x_0) \sim D_1^{-1}$ (see panels (a) and (b) of Figs. 2 and 3).

In Fig. 2c a density plot of $\tau_{MRT}(x_0)/\tau_d(x_0)$ versus $L_1$ and $D_1$ is shown. The white dotted line marks the position of the NES maxima.

The maxima and all curves increase as the value of the left boundary $L_1$ decreases. This gives rise to an increasing size of the basin of attraction of the metastable state [20,21], responsible for the increase in the normalized MRT. Furthermore, in the limit $D_1 \to 0$ there are three different asymptotic values of the normalized MRT, the value of which increases with the size of the basin of attraction when $L_1$ varies from 0 to $-6$ (see the next section and Appendix A, paragraph 1). We note that in the limit $D_1 \to 0$, and for unstable initial position of the particle, there is a divergent behavior of $\tau_{MRT}(x_0)$ with a Gaussian noise source [7–13, 15]. For Lévy flights, however, $\tau_{MRT}(x_0)$ exhibits a finite, nonmonotonic behavior as a function of the noise intensity parameter $D_1$, with finite asymptotic values in the limit $D_1 \to 0$. Due to the heavy tails of the distribution, a particle spends a finite amount of time in the metastable area even in the limit $D_1 \to 0$. For very large noise intensity parameter, in the limit $D_1 \to \infty$, the normalized MRT follows a power-law behavior as a function of the noise intensity parameter [3,4].

In Fig. 3, the MRT $\tau_{MRT}(x_0)$ versus $D_1$ in a semilog plot for different positions $L_2$ of the right boundary at a fixed value of the left boundary $L_1 = -4$ is shown. The different values of $L_2$ range from 3 to 7 with steps 0.2. Again, a full nonmonotonic behavior of $\tau_{MRT}(x_0)$ versus $D_1$ for all values of $L_2$ investigated is observed, with different asymptotic values of the MRT for $D_1 \to 0$. Panel (b) of Fig.3 shows the results obtained using the same parameter values, with numerical simulations of Eq. (1). The agreement with the theoretical exact results of Eq. (29) is excellent. In Fig. 3c a density plot of $\tau_{MRT}(x_0)$ versus $L_2$ and $D_1$ is shown. The white dotted line marks the position of the NES maxima.

**Asymptotic behaviors**   The asymptotic behaviors shown in Figs. 2 and 3 reproduce the asymptotic expressions of Eq. (29) in the limits $D_1 \to 0$ and $D_1 \to \infty$. In particular, for $D_1 \to 0$ we have (see Appendix A, paragraph 1)

$$\tau_{MRT}(x_0) \simeq C_1 \left[ \frac{m}{x_0 - m} + \frac{1}{2} \ln \frac{x_0 - m}{x_0 + m} \right] + \tau_d(x_0), \tag{33}$$

where

$$C_1 \simeq \begin{cases} 1, & L_1 < -m, \\ 1/2, & L_1 = -m, \\ 0, & L_1 > -m, \end{cases} \tag{34}$$

and $x = -m$ is the position of a potential well (see Fig. 1), giving rise to three different asymptotic values of the MRT for $D_1 \to 0$ in the case of fixed $L_2$ (see Fig. 2) and different asymptotic values depending on the different values of $L_2$ in the case of fixed $L_1$ (see Fig. 3).

---

[2]In particular see the Refs. [74] and [75] of Ref. [43].

For $D_1 \to \infty$ (see Appendix A, paragraph 2) we have

$$\tau_{MRT}(x_0) \sim \frac{1}{D_1}, \tag{35}$$

that is a power law behavior of the MRT as a function of the noise intensity parameter (see Figs. 2 and 3).

## 4 Conclusions

We obtain the general equations useful to calculate the MRT for superdiffusion in the form of symmetric Lévy flights, for an arbitrary Lévy index $\alpha$ and an arbitrary smooth potential profile with a sink. For a Cauchy-driven noise ($\alpha = 1$) we find the closed expression in quadratures of the MRT as a function of the noise intensity parameter, the initial position, and the parameters of the potential. The interplay between trapping in the metastable state, at small noise intensities, and long jumps of Lévy flights produces a finite nonmonotonic enhancement of the mean residence time in the metastable state. Our general equations serve as a valuable tool for describing diverse dynamical behaviors in complex systems, particularly those characterized by anomalous diffusion and non-exponential relaxation phenomena, such as spatially extended systems [45].

## Acknowledgments

**Funding information** This work was partially supported by Italian Ministry of University and Research (MUR) and National Center for Physics and Mathematics (section No. 9 of scientific program "Artificial intelligence and big data in technical, industrial, natural and social systems").

**Author contributions** AAD: Conceptualization; Formal analysis; Investigation; Methodology; Validation; Visualization; Writing - original draft; Writing - review & editing. CG: Data curation; Formal analysis; Investigation; Methodology; Software; Validation; Visualization; Writing - review & editing. BS: Conceptualization; Formal analysis; Investigation; Methodology; Validation; Visualization; Writing - original draft; Writing - review & editing.

## A Investigation of the MRT in the limits $D_1 \to 0$ and $D_1 \to \infty$

To investigate the MRT in the asymptotic limits $D_1 \to 0$ and $D_1 \to \infty$, we start from the expressions of the parameters $\lambda_1$ and $\lambda_2$ of Eq. (22)

$$
\begin{aligned}
\lambda_1 &= \left(m^4 + D_1^2\right)^{1/4} \sin\left[\frac{1}{2} \arctan\left(\frac{D_1}{m^2}\right)\right], \\
\lambda_2 &= \left(m^4 + D_1^2\right)^{1/4} \cos\left[\frac{1}{2} \arctan\left(\frac{D_1}{m^2}\right)\right],
\end{aligned}
\tag{A.1}
$$

and, using trigonometry formulas, we rewrite Eq. (A.1) in a simpler form

$$\lambda_1 = \frac{m}{\sqrt{2}} \sqrt{\sqrt{1 + \frac{D_1^2}{m^4}} - 1},$$

$$\lambda_2 = \frac{m}{\sqrt{2}} \sqrt{\sqrt{1 + \frac{D_1^2}{m^4}} + 1}. \tag{A.2}$$

### 1. Asymptotics for $D_1 \to 0$

For small values of $D_1$, we can use the approximate expansion: $\sqrt{1+x} \simeq 1 + x/2 - x^2/8$, where $x = D_1^2/L_1^4 \ll 1$. As a result, we obtain

$$\lambda_1 \simeq \frac{D_1}{2m}\left(1 - \frac{D_1^2}{8m^4}\right), \qquad \lambda_2 \simeq m\left(1 + \frac{D_1^2}{8m^4}\right). \tag{A.3}$$

First of all, it is better to write the expression for $A$ in another form. Using the well-known relation

$$\arctan\frac{1}{x} = \frac{\pi}{2} - \text{arccot}\frac{1}{x} = \frac{\pi}{2} - \arctan x,$$

we can rewrite the expressions (29) for $A$ and $B$ in the following form

$$A = \arctan\frac{\lambda_1}{\lambda_2 + L_1} - \arctan\frac{\lambda_1}{\lambda_2 + L_2} + \pi \cdot 1(-\lambda_2 - L_1),$$

$$B = \frac{1}{2}\ln\frac{\lambda_1^2 + (L_2 + \lambda_2)^2}{\lambda_1^2 + (L_1 + \lambda_2)^2}, \tag{A.4}$$

where $1(x)$ is the step function.

Substituting Eq. (A.3) into Eq. (A.4) for $A$ and $B$ we have

$$A \simeq \begin{cases} \pi + D_1(L_2 - L_1)/[2m(m + L_1)(m + L_2)], & L_1 < -m, \\ \pi/2 - D_1(3m + L_2)/[4m^2(m + L_2)], & L_1 = -m, \\ D_1(L_2 - L_1)/[2m(m + L_1)(m + L_2)], & L_1 > -m. \end{cases} \tag{A.5}$$

$$B \simeq \begin{cases} \ln[(m + L_2)/|m + L_1|], & L_1 \neq -m, \\ \ln[2m(m + L_2)/D_1], & L_1 = -m. \end{cases} \tag{A.6}$$

As seen from Eq. (A.5), the value $A$ does not go to zero in the limit $D_1 \to 0$ in the case $L_1 \leq -m$.

Substitution of Eqs. (A.5) and (A.6) into Eq. (29) gives in the limit $D_1 \to 0$

$$\tau_{MRT}(x_0) \simeq \frac{2mC_1}{\pi}\int_{x_0}^{\infty}\frac{dz}{(z - m)^2(z + m)} + \tau_d(x_0), \tag{A.7}$$

where

$$C_1 \simeq \begin{cases} \pi, & L_1 < -m, \\ \pi/2, & L_1 = -m, \\ 0, & L_1 > -m, \end{cases} \tag{A.8}$$

and $\tau_d(x_0)$ is the dynamical time.

The integral in Eq. (A.7) can be calculated in the analytical form. As a result, we obtain in the limit $D_1 \to 0$

$$\tau_{MRT}(x_0) \simeq C_1\left[\frac{m}{x_0 - m} + \frac{1}{2}\ln\frac{x_0 - m}{x_0 + m}\right] + \frac{1}{2m}\ln\frac{(L_2 - m)(x_0 + m)}{(L_2 + m)(x_0 - m)}. \tag{A.9}$$

This gives rise to three different asymptotic values of the MRT for $D_1 \to 0$ in the case of fixed $L_2$ (see Fig. 2)and different asymptotic values depending on the different values of $L_2$ in the case of fixed $L_1$ (see Fig. 3).

**2. Asymptotics for $D_1 \to \infty$**

Now we consider the case of very large $D_1$. From Eq. (A.2) we easily find

$$\lambda_1 \simeq \sqrt{\frac{D_1}{2}}, \qquad \lambda_2 \simeq \sqrt{\frac{D_1}{2}}. \tag{A.10}$$

Substituting Eq. (A.10) into Eqs. (30) we obtain the following approximate expressions for the constants $A$ and $B$

$$A = B \simeq \frac{L_2 - L_1}{\sqrt{2D_1}}. \tag{A.11}$$

Substitution of Eqs. (A.10) and (A.11) into Eq. (29) gives

$$\tau_{MRT}(x_0) \simeq \frac{L_2 - L_1}{\pi} \int_{x_0}^{\infty} \frac{(z + \sqrt{2D_1})\,dz}{(z^2 - m^2)^2 + D_1^2} \tag{A.12}$$

$$+ \frac{D_1}{\pi} \int_{x_0}^{\infty} \ln\left|\frac{z - L_1}{z - L_2}\right| \frac{dz}{(z^2 - m^2)^2 + D_1^2} + \int_{x_0}^{L_2} \frac{(z^2 - m^2)\,dz}{(z^2 - m^2)^2 + D_1^2}.$$

The first integral in Eq. (A.12) can be calculated analytically and for the large $D_1$ gives

$$\frac{L_2 - L_1}{\pi} \int_{x_0}^{\infty} \frac{(z + \sqrt{2D_1})\,dz}{(z^2 - m^2)^2 + D_1^2} \simeq \frac{3(L_2 - L_1)}{4D_1}. \tag{A.13}$$

The second integral in Eq. (A.12) can be estimated for large $D_1$ using the mean value theorem for a definite integral, namely

$$\frac{D_1}{\pi} \int_{x_0}^{\infty} \ln\left|\frac{z - L_1}{z - L_2}\right| \frac{dz}{(z^2 - m^2)^2 + D_1^2}$$

$$\simeq \frac{D_1}{\pi} \ln\left(\frac{\sqrt{D_1} - L_1}{\sqrt{D_1} - L_2}\right) \int_{x_0}^{\infty} \frac{dz}{(z^2 - m^2)^2 + D_1^2}$$

$$\simeq \frac{(L_2 - L_1)\sqrt{D_1}}{\pi} \int_{x_0}^{\infty} \frac{dz}{(z^2 - m^2)^2 + D_1^2} \simeq \frac{(L_2 - L_1)\sqrt{2}}{4D_1} \sim \frac{1}{D_1}.$$

The last integral in Eq. (A.12), due to the finite limits, can be easily estimated

$$\int_{x_0}^{L_2} \frac{(z^2 - m^2)\,dz}{(z^2 - m^2)^2 + D_1^2} \simeq \frac{1}{D_1^2}\left[\frac{L_2^3 - x_0^3}{3} - m^2(L_2 - x_0)\right] \sim \frac{1}{D_1^2}. \tag{A.14}$$

Taking into account Eqs. (A.13), (A.14), and (A.14), we find finally for large $D_1$

$$\tau_{MRT}(x_0) \sim \frac{1}{D_1}, \tag{A.15}$$

that is a power-law behavior in agreement with previous investigations (e.g., Ref. [3]) and numerical simulations shown in Figs. 2 and 3.

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
