# Peer review of "Enhancement of stability of metastable states in the presence of Lévy noise"

_SciPost Physics, doi:SciPost Phys. 18, 006 (2025)_

## Round 1 · Referee Report · Anonymous (Referee 1) · 2024-6-4

Strengths

1. First exact analytical derivation of NLRT with Levy noise.
2. Demonstration of significant Noise Enhanced Stabilization effect for Levy noise case.

Weaknesses

1. Missing figure plots
2. Lack of proper citations of origial papers, describing basic definitions.

Report

This manuscript presents the results of analytical derivation of a Nonlinear Relaxation Time (NLRT) in the case of Levy noise. In particular, the following important results are derived: the exact results of the NLRT for a particle moving in an arbitrary potential profile in the presence of Lévy noise; a closed expression written by quadratures for NLRT for the particular case of Cauchy noise in cubic metastable potential. To my knowledge, these are the first exact analytical results for any more complex than Markovian case, which have numerous applications in physics, but especially in biology. Besides that, using the obtained analytical results the authors investigate the Noise Enhanced Stabilization effect and show its significant amplification in comparison with white noise case. The paper is interesting and clearly written, so it can be recommended for publication after addressing minor comments listed below.

Comments:
1. In Fig. 3b the only empty frame without any curves is visible, please correct.
2. When describing the basic definition for NLRT (3),(4), the authors should cite the first paper
https://doi.org/10.1016/0378-4371(95)00395-9, where this definition was used for exact derivation of NLRT of a Brownian particle in a smooth potential. Later, this definition was generalized to arbitrary moments of transition times https://doi.org/10.1016/S0375-9601(97)00599-9. Also, it has been demonstrated there for smooth symmetric potentials, that the moments of the First Passage times to the point of symmetry completely coincide with the corresponding moments of transition times. As suggestion for future studies, the authors may try to extend the obtained results to the case of the standard deviation of transition time, which has important applications for description of switching errors of various electronic devices. Another important paper, where the NLRT for the first time has been expressed by quadratures for Markovian processes in smooth potentials, is https://doi.org/10.1016/0921-4534(96)00426-1. There, the NES effect, studied by the authors, was described analytically both using the exact expression and asymptotic series in the low noise limit, see the plots in Fig. 4, so this reference should be added to the list of papers, devoted to NES effect.

Requested changes

1. In Fig. 3b the only empty frame without any curves is visible, please correct.
2. When describing the basic definition for NLRT (3),(4), the authors should cite the first paper
https://doi.org/10.1016/0378-4371(95)00395-9, where this definition was used for exact derivation of NLRT of a Brownian particle in a smooth potential. Later, this definition was generalized to arbitrary moments of transition times https://doi.org/10.1016/S0375-9601(97)00599-9. Also, it has been demonstrated there for smooth symmetric potentials, that the moments of the First Passage times to the point of symmetry completely coincide with the corresponding moments of transition times. As suggestion for future studies, the authors may try to extend the obtained results to the case of the standard deviation of transition time, which has important applications for description of switching errors of various electronic devices. Another important paper, where the NLRT for the first time has been expressed by quadratures for Markovian processes in smooth potentials, is https://doi.org/10.1016/0921-4534(96)00426-1. There, the NES effect, studied by the authors, was described analytically both using the exact expression and asymptotic series in the low noise limit, see the plots in Fig. 4, so this reference should be added to the list of papers, devoted to NES effect.

Recommendation

Ask for minor revision

  • validity: top
  • significance: high
  • originality: top
  • clarity: high
  • formatting: excellent
  • grammar: excellent

Author:  Bernardo Spagnolo  on 2024-06-15  [id 4569]

(in reply to Report 1 on 2024-06-04)

The Referee writes: 1 – Objection “1. In Fig. 3b the only empty frame without any curves is visible, please correct.”

Our response: 1 - Reply to objection 1. Fig. 3b has been corrected, now all curves are visible in the revised version.

The Referee writes: 2 - Objection 2. When describing the basic definition for NLRT (3),(4), the authors should cite the first paper https://doi.org/10.1016/0378-4371(95)00395-9, where this definition was used for exact derivation of NLRT of a Brownian particle in a smooth potential. Later, this definition was generalized to arbitrary moments of transition times https://doi.org/10.1016/S0375-9601(97)00599-9. Also, it has been demonstrated there for smooth symmetric potentials, that the moments of the First Passage times to the point of symmetry completely coincide with the corresponding moments of transition times. As suggestion for future studies, the authors may try to extend the obtained results to the case of the standard deviation of transition time, which has important applications for description of switching errors of various electronic devices. Another important paper, where the NLRT for the first time has been expressed by quadratures for Markovian processes in smooth potentials, is https://doi.org/10.1016/0921-4534(96)00426-1. There, the NES effect, studied by the authors, was described analytically both using the exact expression and asymptotic series in the low noise limit, see the plots in Fig. 4, so this reference should be added to the list of papers, devoted to NES effect.

Our response: 2 - Reply to objection We thank the Referee for suggesting relevant references that we will add in the revised version of the manuscript with related comments. In particular in the introduction of the revised version we will include the following references: i) K. Binder, “Time-Dependent Ginzburg-Landau Theory of Nonequilibrium Relaxation”, Phys. Rev. B 8, 3423 (1973), https://doi.org/10.1103/PhysRevB.8.3423; ii) N. V. Agudov and A. N. Malakhov, “Nonstationary diffusion through arbitrary piecewise-linear potential profile. Exact solution and time characteristics”, Radiophys. Quantum Electron. 36, 97 (1993), https://doi.org/10.1007/BF01059491; iii) A.N. Malakhov, A.L. Pankratov, “Exact solution of Kramers' problem for piecewise parabolic potential profiles”, Physica A: Statistical Mechanics and its Applications 229, 109-126 (1996), https://doi.org/10.1016/0378-4371(95)00395-9; iv) A.L. Pankratov, “On certain time characteristics of dynamical systems driven by noise”, Physics Letters A 234, 329-335 (1997), https://doi.org/10.1016/S0375-9601(97)00599-9; v) A.N. Malakhov, A.L. Pankratov, “Influence of thermal fluctuations on time characteristics of a single Josephson element with high damping exact solution”, Physica C: Superconductivity 269, 46-54 (1996), https://doi.org/10.1016/0921-4534(96)00426-1.

Finally, the changes requested by the Referee coincide with the Comments, to which we have already responded, see above. Specifically, the list of changes is as follows: 1 – New Fig. 3b 2 – Five new references reported in the above response to objection 2. 3 – Related comments on these new references in the introduction of the revised manuscript.

Attachment:

Reply_to_Report_1_15.06.24.pdf

---

## Round 1 · Referee Report · Anonymous (Referee 2) · 2024-6-13

Strengths

1. Obtained an exact analytical result for the nonlinear relaxation time
2. Found non-monotonic behaviour interpreted as enhanced stability of metastable states

Weaknesses

1. Literature review in introduction is too generic
2. Numerical confirmation missing

Report

The authors investigate the nonlinear relaxation time (NLRT) of a particle in a metastable potential driven by Levy noise. An exact expression for the NLRT is derived and further evaluated analytically for the special case of a cubic metastable potential. The solution reveals the interesting phenomenon of noise enhanced stability, previously observed for Gaussian noise.

The study is carefully performed and mathematically sound. I am quite intrigued by the fact that such an exact solution for the NLRT can be found, in particular since other studies mostly consider asymptotic treatments of similar escape problems for metastable systems. The manuscript is thus certainly suitable for publication in principle, but I recommend that the authors consider the following comments to put the work better into context as well as provide independent confirmation.

1. The introduction on the problem is too generic. The problem of escape from metastable states driven by Levy-type noise has attracted a considerable amount of theoretical research over the last two decades. The authors write that "there are a lot of numerical results and some analytical approximations [3,32,35]." But there are lot more than 3 articles that are relevant here (see, e.g., the references discussed in Ref.[35]). This does not have to be exhaustive, but it would be useful to know the different technical approaches used previously and the results found.

2. Likewise, I wonder if the discussion in Section 2 requires some more references. Has this approach been discussed previously for Gaussian noise?

3. The discussion of the ratio <\tau_NLRT>/\tau_d below Eqs.(28,29) is confusing, because \tau_d appears without any proper introduction. It would be clearer if first the behaviour of <\tau_NLRT> would be discussed by itself together with the relevant figures (Fig.2b and Figs.3a,b). Then, \tau_d could be introduced (this needs more explanations) and the ratio <\tau_NLRT>/\tau_d discussed.

4. The authors should clarify whether "nonlinear relaxation time" and "mean residence time" are the same quantity. If they are identical, I would recommend to use only one of the two terminologies throughout the manuscript.

5. The analytical results should be supplemented by numerical results. It is straightforward to simulate Eq.(2), thus numerical confirmation should be provided.

6. The authors mention that the enhancement of stability of metastable states has been observed previously for Gaussian noise. How does \tau_NLRT differ quantitatively from this case? It would be important to understand how the stability is affected by the non-Gaussianity of the noise.

Requested changes

See report

Recommendation

Ask for minor revision

  • validity: good
  • significance: ok
  • originality: ok
  • clarity: high
  • formatting: good
  • grammar: acceptable

Author:  Bernardo Spagnolo  on 2024-09-20  [id 4790]

(in reply to Report 2 on 2024-06-13)
Category:
objection
reply to objection

The Referee writes: 1 – Objection “1. The introduction on the problem is too generic. The problem of escape from metastable states driven by Levy-type noise has attracted a considerable amount of theoretical research over the last two decades. The authors write that “there are a lot of numerical results and some analytical approximations [3,32,35].” But there are lot more than 3 articles that are relevant here (see, e.g., the references discussed in Ref. [35]). This does not have to be exhaustive, but it would be useful to know the different technical approaches used previously and the results found.”

Our response: 1 - Reply to objection 1. The Referee is right. We have thoroughly rewritten the introduction to place the manuscript's subject in a broader and more scientifically appropriate context, incorporating additional relevant material. Specifically, we have emphasized the key question under investigation. Additionally, we provide a brief review of theoretical studies on the problem of escape from metastable states driven by Lévy noise, published over the past two decades. This review includes extensive research conducted through both numerical simulations and analytical approximations, with relevant citations (see references [3, 41, 44, 46-55]). Additionally, we have revised and improved the abstract for greater clarity and precision.

The Referee writes: 2 - Objection 2. “Likewise, I wonder if the discussion in Section 2 requires some more references. Has this approach been discussed previously for Gaussian noise?”

Our response: 2 - Reply to objection The referee is correct. We have added new references and properly contextualized them. As mentioned in the introduction and immediately after equation (1), our study applies to the stability index 𝛼 of the Lévy distribution within the range 0 < 𝛼 < 2. In other words, it does not apply to 𝛼 =2, which corresponds to Gaussian noise. The phenomenon of noise-enhanced stability under Gaussian noise has been thoroughly investigated using different approaches, as highlighted in the new references [7-15], particularly through studies on the ordinary Fokker-Planck equation and, in some cases, using functional analysis.

The Referee writes: 3 - Objection 3. “The discussion of the ratio ⟨τNLRT⟩/τd below Eqs. (28,29) is confusing, because τd appears without any proper introduction. It would be clearer if first the behaviour of ⟨τNLRT⟩ would be discussed by itself together with the relevant figures (Fig.2b and Figs. 3a,b). Then, τd could be introduced (this needs more explanations) and the ratio ⟨τNLRT⟩/τd discussed.”

Our response: 3 - Reply to objection The referee is correct. First we changed τNLRT to 𝜏𝑀𝑅𝑇. We have now properly introduced the dynamic time 𝜏𝑑 immediately after presenting the exact quadrature result in Eq. (29), along with the necessary explanations. Additionally, in the footnote on page 7, we have described the non-normalized behavior of the mean residence time in the metastable state, 𝜏𝑀𝑅𝑇(x0), as a function of the noise intensity parameter 𝐷1 with fixed 𝐿2. This shows the same non-monotonic behavior, including a maximum, though with different scaling on the vertical axis of Fig. 2. In Fig. 3, the MRT 𝜏𝑀𝑅𝑇(x0) versus 𝐷1, with fixed L1, is shown. Additionally, we have added more physical insights on the new Figs. 2 and 3, along with details on the numerical integration of the Langevin equation (1). Please refer to the updated page 8 of the revised manuscript.

The Referee writes: 4 - Objection 4. “The authors should clarify whether "nonlinear relaxation time" and "mean residence time" are the same quantity. If they are identical, I would recommend to use only one of the two terminologies throughout the manuscript.”

Our response: 4 - Reply to objection The referee is correct. We have consistently used the term “mean residence time” throughout the revised manuscript.

The Referee writes: 5 - Objection 5. “The analytical results should be supplemented by numerical results. It is straightforward to simulate Eq.(2), thus numerical confirmation should be provided.”

Our response: 5 - Reply to objection We thank the Referee for the suggestion, which we have implemented by complementing the analytical results with numerical simulations of Eq. (1). We found excellent agreement between the exact theoretical results of Eq. (29) and the numerical simulations of Eq. (1), as demonstrated in Figs. 2b and 3b, thereby providing strong numerical confirmation.

The Referee writes: 6 - Objection 6. “The authors mention that the enhancement of stability of metastable states has been observed previously for Gaussian noise. How does 𝜏NL𝑅𝑇 differ quantitatively from this case? It would be important to understand how the stability is affected by the non-Gaussianity of the noise.”

Our response: 6 - Reply to objection First we changed τNLRT to 𝜏𝑀𝑅𝑇. Then, on page 8 we described how 𝜏𝑀𝑅𝑇 with non-Gaussian noise differs qualitatively and quantitatively from the case of Gaussian noise. In particular, we note that in the limit D1 →0, and for unstable initial position of the particle, there is a divergent behavior of τMRT(x0) with a Gaussian noise source, see Refs. [7–13, 15]. For Lévy flights, however, τMRT (x0) exhibits a finite, nonmonotonic behavior as a function of the noise intensity parameter D1, with finite asymptotic values in the limit D1 →0. Due to the heavy tails of the distribution, a particle spends a finite amount of time in the metastable area even in the limit D1 → 0. For very large noise intensity parameter, in the limit D1 →∞, the normalized MRT follows a power-law behavior as a function of the noise intensity parameter, see Refs. [3,4].

Attachment:

Reply_to_Report_2_19.09.24.pdf

Author:  Bernardo Spagnolo  on 2024-09-20  [id 4789]

(in reply to Report 2 on 2024-06-13)
Category:
objection
reply to objection

The Referee writes: 1 – Objection “1. The introduction on the problem is too generic. The problem of escape from metastable states driven by Levy-type noise has attracted a considerable amount of theoretical research over the last two decades. The authors write that “there are a lot of numerical results and some analytical approximations [3,32,35].” But there are lot more than 3 articles that are relevant here (see, e.g., the references discussed in Ref. [35]). This does not have to be exhaustive, but it would be useful to know the different technical approaches used previously and the results found.”

Our response: 1 - Reply to objection 1. The Referee is right. We have thoroughly rewritten the introduction to place the manuscript's subject in a broader and more scientifically appropriate context, incorporating additional relevant material. Specifically, we have emphasized the key question under investigation. Additionally, we provide a brief review of theoretical studies on the problem of escape from metastable states driven by Lévy noise, published over the past two decades. This review includes extensive research conducted through both numerical simulations and analytical approximations, with relevant citations (see references [3, 41, 44, 46-55]). Additionally, we have revised and improved the abstract for greater clarity and precision.

The Referee writes: 2 - Objection 2. “Likewise, I wonder if the discussion in Section 2 requires some more references. Has this approach been discussed previously for Gaussian noise?”

Our response: 2 - Reply to objection The referee is correct. We have added new references and properly contextualized them. As mentioned in the introduction and immediately after equation (1), our study applies to the stability index 𝛼 of the Lévy distribution within the range 0 < 𝛼 < 2. In other words, it does not apply to 𝛼 =2, which corresponds to Gaussian noise. The phenomenon of noise-enhanced stability under Gaussian noise has been thoroughly investigated using different approaches, as highlighted in the new references [7-15], particularly through studies on the ordinary Fokker-Planck equation and, in some cases, using functional analysis.

The Referee writes: 3 - Objection 3. “The discussion of the ratio ⟨τNLRT⟩/τd below Eqs. (28,29) is confusing, because τd appears without any proper introduction. It would be clearer if first the behaviour of ⟨τNLRT⟩ would be discussed by itself together with the relevant figures (Fig.2b and Figs. 3a,b). Then, τd could be introduced (this needs more explanations) and the ratio ⟨τNLRT⟩/τd discussed.”

Our response: 3 - Reply to objection The referee is correct. First we changed τNLRT to 𝜏𝑀𝑅𝑇. We have now properly introduced the dynamic time 𝜏𝑑 immediately after presenting the exact quadrature result in Eq. (29), along with the necessary explanations. Additionally, in the footnote on page 7, we have described the non-normalized behavior of the mean residence time in the metastable state, 𝜏𝑀𝑅𝑇(x0), as a function of the noise intensity parameter 𝐷1 with fixed 𝐿2. This shows the same non-monotonic behavior, including a maximum, though with different scaling on the vertical axis of Fig. 2. In Fig. 3, the MRT 𝜏𝑀𝑅𝑇(x0) versus 𝐷1, with fixed L1, is shown. Additionally, we have added more physical insights on the new Figs. 2 and 3, along with details on the numerical integration of the Langevin equation (1). Please refer to the updated page 8 of the revised manuscript.

The Referee writes: 4 - Objection 4. “The authors should clarify whether "nonlinear relaxation time" and "mean residence time" are the same quantity. If they are identical, I would recommend to use only one of the two terminologies throughout the manuscript.”

Our response: 4 - Reply to objection The referee is correct. We have consistently used the term “mean residence time” throughout the revised manuscript.

The Referee writes: 5 - Objection 5. “The analytical results should be supplemented by numerical results. It is straightforward to simulate Eq.(2), thus numerical confirmation should be provided.”

Our response: 5 - Reply to objection We thank the Referee for the suggestion, which we have implemented by complementing the analytical results with numerical simulations of Eq. (1). We found excellent agreement between the exact theoretical results of Eq. (29) and the numerical simulations of Eq. (1), as demonstrated in Figs. 2b and 3b, thereby providing strong numerical confirmation.

The Referee writes: 6 - Objection 6. “The authors mention that the enhancement of stability of metastable states has been observed previously for Gaussian noise. How does 𝜏NL𝑅𝑇 differ quantitatively from this case? It would be important to understand how the stability is affected by the non-Gaussianity of the noise.”

Our response: 6 - Reply to objection First we changed τNLRT to 𝜏𝑀𝑅𝑇. Then, on page 8 we described how 𝜏𝑀𝑅𝑇 with non-Gaussian noise differs qualitatively and quantitatively from the case of Gaussian noise. In particular, we note that in the limit D1 →0, and for unstable initial position of the particle, there is a divergent behavior of τMRT(x0) with a Gaussian noise source, see Refs. [7–13, 15]. For Lévy flights, however, τMRT (x0) exhibits a finite, nonmonotonic behavior as a function of the noise intensity parameter D1, with finite asymptotic values in the limit D1 →0. Due to the heavy tails of the distribution, a particle spends a finite amount of time in the metastable area even in the limit D1 → 0. For very large noise intensity parameter, in the limit D1 →∞, the normalized MRT follows a power-law behavior as a function of the noise intensity parameter, see Refs. [3,4].

Attachment:

Reply_to_Report_2.pdf

---

## Round 2 · Referee Report · Adrian Baule (Referee 3) · 2024-10-30

Report
The authors have addressed all my previous comments in detail, considerably improving the manuscript. In particular, the additional numerical results show excellent agreement between simulations and theory corroborating these important results. I recommend publication of the manuscript without any further changes.
Recommendation
Publish (easily meets expectations and criteria for this Journal; among top 50%)

---

## Round 2 · Author Response

Dear Editor,
We are resubmitting the revised version of our manuscript titled "Enhancement of stability of metastable states in the presence of Lévy noise" for consideration in SciPost. We believe that we have thoroughly addressed all the questions, comments, and criticisms raised by both Referee 1 and 2 in their reports.
We are confident that the revisions have strengthened our manuscript and we consider it now suitable for publication in your journal.
Thank you for your consideration.
On behalf of all authors
Bernardo Spagnolo

---

## Round 2 · List of Changes

Concerning Referee 1
Comments
The Referee writes:
1 – Objection
“1. In Fig. 3b the only empty frame without any curves is visible, please correct.”
Our response:
1 - Reply to objection
1. Fig. 3b has been corrected, now all curves are visible in the revised version.
Change: Fig. 3b is now new.
The Referee writes:
2 - Objection
2. When describing the basic definition for NLRT (3),(4), the authors should cite the first paper https://doi.org/10.1016/0378-4371(95)00395-9, where this definition was used for exact derivation of NLRT of a Brownian particle in a smooth potential. Later, this definition was generalized to arbitrary moments of transition times https://doi.org/10.1016/S0375-9601(97)00599-9. Also, it has been demonstrated there for smooth symmetric potentials, that the moments of the First Passage times to the point of symmetry completely coincide with the corresponding moments of transition times. As suggestion for future studies, the authors may try to extend the obtained results to the case of the standard deviation of transition time, which has important applications for description of switching errors of various electronic devices. Another important paper, where the NLRT for the first time has been expressed by quadratures for Markovian processes in smooth potentials, is https://doi.org/10.1016/0921-4534(96)00426-1. There, the NES effect, studied by the authors, was described analytically both using the exact expression and asymptotic series in the low noise limit, see the plots in Fig. 4, so this reference should be added to the list of papers, devoted to NES effect.
Our response:
2 - Reply to objection
We thank the Referee for suggesting relevant references that we will add in the revised version of the manuscript with related comments. In particular in the introduction of the revised version we will include the following references: i) K. Binder, “Time-Dependent Ginzburg-Landau Theory of Nonequilibrium Relaxation”, Phys. Rev. B 8, 3423 (1973),
https://doi.org/10.1103/PhysRevB.8.3423; ii) N. V. Agudov and A. N. Malakhov, “Nonstationary diffusion through arbitrary piecewise-linear potential profile. Exact solution and time characteristics”, Radiophys. Quantum Electron. 36, 97 (1993), https://doi.org/10.1007/BF01059491; iii) A.N. Malakhov, A.L. Pankratov, “Exact solution of Kramers' problem for piecewise parabolic potential profiles”, Physica A: Statistical Mechanics and its Applications 229, 109-126 (1996), https://doi.org/10.1016/0378-4371(95)00395-9; iv) A.L. Pankratov, “On certain time characteristics of dynamical systems driven by noise”, Physics Letters A 234, 329-335 (1997), https://doi.org/10.1016/S0375-9601(97)00599-9; v) A.N. Malakhov, A.L. Pankratov, “Influence of thermal fluctuations on time characteristics of a single Josephson element with high damping exact solution”, Physica C: Superconductivity 269, 46-54 (1996), https://doi.org/10.1016/0921-4534(96)00426-1.
Change: New references added: Refs. [61-66]. New sentences on page 4, after Eq. (5).
Finally, the changes requested by the Referee coincide with the Comments, to which we have already responded, see above. Specifically, the list of changes is as follows:
1 – New Fig. 3b
2 – Five new references reported in the above response to objection 2.
3 - New sentences on page 4, after Eq. (5).
Concerning Referee 2
The Referee writes:
1 – Objection
“1. The introduction on the problem is too generic. The problem of escape from metastable states driven by Levy-type noise has attracted a considerable amount of theoretical research over the last two decades. The authors write that “there are a lot of numerical results and some analytical approximations [3,32,35].” But there are lot more than 3 articles that are relevant here (see, e.g., the references discussed in Ref. [35]). This does not have to be exhaustive, but it would be useful to know the different technical approaches used previously and the results found.”
Our response:
1 - Reply to objection
1. The Referee is right. We have thoroughly rewritten the introduction to place the manuscript's subject in a broader and more scientifically appropriate context, incorporating additional relevant material. Specifically, we have emphasized the key question under investigation. Additionally, we provide a brief review of theoretical studies on the problem of escape from metastable states driven by Lévy noise, published over the past two decades. This review includes extensive research conducted through both numerical simulations and analytical approximations, with relevant citations (see references [3, 41, 44, 46-55]). Additionally, we have revised and improved the abstract for greater clarity and precision.
Change: New introduction, new abstract and new references added [41, 44, 46-55].
The Referee writes:
2 - Objection
2. “Likewise, I wonder if the discussion in Section 2 requires some more references. Has this approach been discussed previously for Gaussian noise?”
Our response:
2 - Reply to objection
The referee is correct. We have added new references and properly contextualized them. As mentioned in the introduction and immediately after equation (1), our study applies to the stability index 𝛼 of the Lévy distribution within the range 0 < 𝛼 < 2. In other words, it does not apply to 𝛼 =2, which corresponds to Gaussian noise. The phenomenon of noise-enhanced stability under Gaussian noise has been thoroughly investigated using different approaches, as highlighted in the new references [7-15], particularly through studies on the ordinary Fokker-Planck equation and, in some cases, using functional analysis.
Change: New references added [56-66].
The Referee writes:
3 - Objection
3. “The discussion of the ratio ⟨τNLRT⟩/τd below Eqs. (28,29) is confusing, because τd appears without any proper introduction. It would be clearer if first the behaviour of ⟨τNLRT⟩ would be discussed by itself together with the relevant figures (Fig.2b and Figs. 3a,b). Then, τd could be introduced (this needs more explanations) and the ratio ⟨τNLRT⟩/τd discussed.”
Our response:
3 - Reply to objection
The referee is correct. First we changed τNLRT to 𝜏𝑀𝑅𝑇. We have now properly introduced the dynamic time 𝜏𝑑 immediately after presenting the exact quadrature result in Eq. (29), along with the necessary explanations. Additionally, in the footnote on page 7, we have described the non-normalized behavior of the mean residence time in the metastable state, 𝜏𝑀𝑅𝑇(x0), as a function of the noise intensity parameter 𝐷1 with fixed 𝐿2. This shows the same non-monotonic behavior, including a maximum, though with different scaling on the vertical axis of Fig. 2. In Fig. 3, the MRT 𝜏𝑀𝑅𝑇(x0) versus 𝐷1, with fixed L1, is shown. Additionally, we have added more physical insights on the new Figs. 2 and 3, along with details on the numerical integration of the Langevin equation (1). Please refer to the updated page 8 of the revised manuscript.
Change: We changed τ_NLRT to 𝜏_𝑀𝑅𝑇. Furthermore, on page 7 we have properly introduced the dynamic time 𝜏𝑑 immediately after presenting the exact quadrature result in Eq. (29), along with the necessary explanations. We added a footnote on page 7, describing the non-normalized behavior of the mean residence time in the metastable state, 𝜏𝑀𝑅𝑇(x0), as a function of the noise intensity parameter 𝐷1 with fixed 𝐿2. Additionally, we have added more physical insights on the new Figs. 2 and 3, along with details on the numerical integration of the Langevin equation (1). Please refer to the updated page 8 of the revised manuscript.
The Referee writes:
4 - Objection
4. “The authors should clarify whether "nonlinear relaxation time" and "mean residence time" are the same quantity. If they are identical, I would recommend to use only one of the two terminologies throughout the manuscript.”
Our response:
4 - Reply to objection
The referee is correct. We have consistently used the term “mean residence time” throughout the revised manuscript.
Change: We have consistently used the term “mean residence time” throughout the revised manuscript.
The Referee writes:
5 - Objection
5. “The analytical results should be supplemented by numerical results. It is straightforward to simulate Eq.(2), thus numerical confirmation should be provided.”
Our response:
5 - Reply to objection
We thank the Referee for the suggestion, which we have implemented by complementing the analytical results with numerical simulations of Eq. (1). We found excellent agreement between the exact theoretical results of Eq. (29) and the numerical simulations of Eq. (1), as demonstrated in Figs. 2b and 3b, thereby providing strong numerical confirmation.
Change: We have implemented by complementing the analytical results with numerical simulations of Eq. (1), see page 8 for details. We show in the new Figs. 2b and 3b the excellent agreement between the exact theoretical results of Eq. (29) and the numerical simulations of Eq. (1).
The Referee writes:
6 - Objection
6. “The authors mention that the enhancement of stability of metastable states has been observed previously for Gaussian noise. How does 𝜏NL𝑅𝑇 differ quantitatively from this case? It would be important to understand how the stability is affected by the non-Gaussianity of the noise.”
Our response:
6 - Reply to objection
First we changed τNLRT to 𝜏𝑀𝑅𝑇. Then, on page 8 we described how 𝜏𝑀𝑅𝑇 with non-Gaussian noise differs qualitatively and quantitatively from the case of Gaussian noise. In particular, we note that in the limit D1 →0, and for unstable initial position of the particle, there is a divergent behavior of τMRT(x0) with a Gaussian noise source, see Refs. [7–13, 15]. For Lévy flights, however, τMRT (x0) exhibits a finite, nonmonotonic behavior as a function of the noise intensity parameter D1, with finite asymptotic values in the limit D1 →0. Due to the heavy tails of the distribution, a particle spends a finite amount of time in the metastable area even in the limit D1 → 0. For very large noise intensity parameter, in the limit D1 →∞, the normalized MRT follows a power-law behavior as a function of the noise intensity parameter, see Refs. [3,4].
Change: We changed τNLRT to 𝜏𝑀𝑅𝑇. On page 8 we described how 𝜏𝑀𝑅𝑇 with non-Gaussian noise differs qualitatively and quantitatively from the case of Gaussian noise.

---

## Editorial Decision

published